# Glucocorticoid Receptor and Ovarian Cancer: From Biology to Therapeutic Intervention

**DOI:** 10.3390/biom13040653

**Published:** 2023-04-05

**Authors:** Roberto Buonaiuto, Giuseppe Neola, Sabrina Chiara Cecere, Aldo Caltavituro, Amedeo Cefaliello, Erica Pietroluongo, Pietro De Placido, Mario Giuliano, Grazia Arpino, Carmine De Angelis

**Affiliations:** 1Department of Clinical Medicine and Surgery, University of Naples Federico II, 80131 Naples, Italy; 2Oncologia Clinica Sperimentale Uro-Ginecologica, Istituto Nazionale Tumori IRCCS Fondazione G Pascale, 80131 Naples, Italy

**Keywords:** ovarian cancer, glucocorticoid receptor, glucocorticoids, chemotherapy resistance

## Abstract

Ovarian cancer (OC) is the leading cause of death from gynecological malignancies worldwide. Fortunately, recent advances in OC biology and the discovery of novel therapeutic targets have led to the development of novel therapeutic agents that may improve the outcome of OC patients. The glucocorticoid receptor (GR) is a ligand-dependent transcriptional factor known for its role in body stress reactions, energy homeostasis and immune regulation. Notably, evidence suggests that GR may play a relevant role in tumor progression and may affect treatment response. In cell culture models, administration of low levels of glucocorticoids (GCs) suppresses OC growth and metastasis. Conversely, high GR expression has been associated with poor prognostic features and long-term outcomes in patients with OC. Moreover, both preclinical and clinical data have shown that GR activation impairs the effectiveness of chemotherapy by inducing the apoptotic pathways and cell differentiation. In this narrative review, we summarize data related to the function and role of GR in OC. To this aim, we reorganized the controversial and fragmented data regarding GR activity in OC and herein describe its potential use as a prognostic and predictive biomarker. Moreover, we explored the interplay between GR and BRCA expression and reviewed the latest therapeutic strategies such as non-selective GR antagonists and selective GR modulators to enhance chemotherapy sensitivity, and to finally provide new treatment options in OC patients.

## 1. Introduction

Ovarian cancer (OC) is the third most common gynecological malignancy worldwide with nearly 314,000 new cases diagnosed in 2020 [1]. Over recent years, the treatment landscape of OC has changed drastically due to advances in the field of tumor biology. However, OC remains the most lethal gynecological tumor with 207,252 deaths recorded globally in 2020 [1]. Notably, high-grade serous ovarian carcinoma (HGSOC), which accounts for nearly 85% of epithelial ovarian carcinomas (EOC), is the most aggressive subtype and is typically diagnosed in the late stages. Surgery and platinum-based chemotherapy have represented the backbone of HGSOC treatment. However, almost 75% of patients develop chemotherapy resistance and eventually relapse [2,3,4]. New therapeutic strategies have been explored, and the introduction of a maintenance regimen with poly-adenosine diphosphate ribose polymerase inhibitors (PARPi) has represented a paradigmatic shift in OC treatment [5,6,7,8]. PARPi has paved the way for a novel targeted-based therapeutic approach, and new potential players driving OC progression and treatment response have rapidly emerged. GR is a nuclear hormone receptor that is mainly known for its role in metabolic homeostasis and stress response [9]. Nevertheless, several studies have highlighted the link between GR activity and tumorigenesis in solid malignancies [10]. In particular, emerging evidence suggests that GR signaling may affect tumor progression and chemotherapy response in patients diagnosed with HGSOC. In this narrative review, we examine the role of GR in OC biology and describe prognostic and predictive values and the therapeutic strategies currently adopted or under development to interfere with GR signaling in OC.

## 2. GR Structure and Function

The glucocorticoid receptor is a ligand-activated transcriptional factor that belongs to the superfamily of nuclear receptors (Figure 1). Nuclear receptors (NRs) are transcriptional factors that, upon ligand-binding, directly interact with DNA responsive regions and promote the recruitment of co-regulator proteins and chromatin remodelers [11,12]. GR is encoded by the nuclear receptor subfamily 3 group C member 1 (*NR3C1*) gene located on chromosome 5 (5q31.3). *NR3C1* consists of ten exons, from 1 to 9β. Exon 1 is an untranslated region. Exon 2 encodes the N-terminal domain (NTD), which is a transactivation domain that is independent of ligand activation. The NTD contains the activation function 1 (AF1) binding site that directly interacts with co-regulator molecules and chromatin-modifiers such as the activator protein 1 (AP1) [13]. Exons 3 and 4 encode the DNA-binding domain (DBD), which contains two zinc finger motifs that are responsible for the recognition of glucocorticoid-responsive elements (GREs) and for GR homodimerization, respectively. Furthermore, exons 5 to 9 encode for the ligand-binding domain (LBD) that directly interacts with glucocorticoids (GCs) and, via the activation function two (AF-2) region, binds coactivator or corepressor proteins [13]. Notably, some GR isoforms are generated through *NR3C1* mRNA alternative splicing, each presenting different structure and transcriptional activities. Among the latter, GRα is the longest isoform and is characterized by an NTD, independent of ligand activation, and is associated with increased sensitivity to GC signaling [14]. Differently, the LBD of GRβ is 35 amino acids shorter than that of GRα, thereby resulting in impaired GC binding [13,14,15]. In particular, GRβ forms transcriptionally inactive heterodimers with GRα and exerts a dominant negative effect on GRα-mediated transactivation. Furthermore, GRβ-mediated signaling can directly ensure or repress the transcription of several genes independent of GRα antagonism. Notably, GRβ activity has been associated with reduced GCs sensitivity, enhanced cancer cell growth and, an induced inflammatory state [16]. In addition, GRγ differs from GRα in terms of a single arginine insertion that occurs in the DBD and that has distinct binding properties. Structural changes decrease GR binding affinity, thus resulting in fluctuating GR activity [13,17]. Moreover, alternative mRNA splicing eventually results in seven NTDs and two C-terminal truncated isoforms with still unclear biological activity [18,19].

The effects of GCs are principally mediated by the GRα isoform. Notably, GRα is physiologically located in the cytoplasm and is complexed with chaperone proteins, such as heat-shock protein (Hsp) 90, Hsp70, p23, and immunophilins FK506 binding protein 51 (FKBP51) and FKBP52 that prevent its degradation and ensure its maturation [20]. Upon GC binding, glycogen synthase kinase 3 (GSK3) and cyclin-dependent kinases phosphorylate the LBD and induce a conformational change to ultimately enhance the interaction with the ATP-dependent motor proteins dyneins, which promote the translocation of GR to the nucleus [10]. Thus, upon nuclear dimerization, GR-DBD binds to specific GREs to promote gene transcription. Alternatively, GR can eventually bind DNA as a monomer, either to GREs or to negative GREs (nGRE), in order to ensure or repress gene transcription, respectively [21]. In addition, GR may indirectly exert its activity by interacting with DNA-binding factors such as AP-1 and the nuclear factor kappa-light-chain-enhancer of activated B cells (NFKB), thereby negatively modulating their transcriptional programs [22].

## 3. GR Physiological and Pathological Functions

NRs physiologically modulate gene expression associated with reproduction, metabolism and inflammatory response [23]. Conversely, aberrant NR-mediated signaling may eventually enhance multiple pathological processes such as carcinogenesis, metabolic disorders and reactive oxygen species production [23,24,25]. Notably GR, upon binding to GCs, is known to drive the stress response by accelerating glucose metabolism and to exert an anti-inflammatory and immune-suppressive function by shaping the adaptive and innate immune responses [26,27]. GR-mediated signaling promotes transrepression of the inflammatory genes NF-kB and AP-1. Notably, GR decreases NF-kB activity by enhancing the expression of the enzyme complex inhibitor of the nuclear factor kappa (IkB) and by repressing the mitogen-activated protein kinase (MAPK)-mediated AP-1 activity [28].

Furthermore, GR-mediated signaling decreases the extravasation of neutrophils toward inflamed tissue by downregulating the expression of cell adhesion molecules on both endothelial cells (e.g., intercellular adhesion molecule 1) and leukocytes (e.g., P-selectin) and reduces chemotactic signaling driven by interleukin-8 (IL-8) secretion [29]. Moreover, GR guides macrophage polarization toward an M2-like phenotype by inhibiting NFkB and AP-1 that synergistically promote an M1-like macrophage status [30]. GR signaling has also been shown to promote T-cell apoptosis by enhancing the activity of proapoptotic family members such as B-cell lymphoma 2 (Bcl-2), interacting mediator of cell death (BIM), and Bcl modifying factor (BMF), as well as by downregulating the antiapoptotic activity of the Bcl-2 protein [10,31]. However, GR can directly modulate T-cell activity regardless of its effects on T-cell apoptosis. Preclinical evidence showed that GR-mediated signaling induced by dexamethasone (DEX) administration increased the expression of programmed death-1 (PD-1) on T-cell and NK-cell surfaces. The increased expression of PD-1 attenuates the T-cell response and promotes cancer immune escape by the interaction between PD-1 and its ligand proteins, namely programmed death ligand-1 (PD-L1) and programmed death ligand-2 (PD-L2) that are expressed on cancer cell surfaces [32]. Similarly, DEX enhances the expression of the cytotoxic T-lymphocyte antigen-4 (CTLA-4) receptor on activated CD4+ and CD8+ T-lymphocytes, thereby promoting T-cell inactivation and immune evasion upon binding of CTLA-4 to its ligands (B7-1 and B7-2) expressed on antigen-presenting cells (APCs) [32].

Increasing evidence suggests that GR plays a role in tumorigenesis beyond the effects it exerts on the immune system. It has also been shown to act either as an oncogene or an oncosuppressor depending on the tumor histotype, and its activity has been explored in several solid tumors [10]. Notably, GR expression has been associated with a favorable prognosis in estrogen-receptor positive (ER+) breast cancer (BC) [33,34,35]. GR signaling interferes with ERα proliferative programs by (i) displacing ERα from DNA-responsive elements, (ii) inhibiting wingless/integrated (Wnt) oncogene proliferative activity, and (iii) reducing the transcription of genes associated with epithelial–mesenchymal transition (EMT) [36,37]. Conversely, GR expression has been associated with poor prognosis, resistance to chemotherapy and increased metastatic potential via activation of phosphatidylinositol 3-kinases (PI3K) signaling in triple negative breast cancer (TNBC) [38,39]. In TNBC preclinical models, increased levels of reactive oxygen species (ROS) and hypoxia-inducible factor (HIF) induce GR phosphorylation that, in turn, promotes the expression of breast tumor kinase (BRK), which is a downstream mediator of multiple growth factor receptors associated with an aggressive and resistant phenotype [40]. Moreover, in patient-derived xenograft (PDX) models, GR activation resulted in increased expression of the transmembrane receptor tyrosine protein kinase (ROR1), which promotes metastatic colonization and reduces cells survival via the activation of the Wnt and hippo pathways [41]. Similarly to BC, the GR duality of action has also been observed in prostate cancer (PC) [10,42,43,44]. In primary hormone-sensitive prostate cancer cell lines, GR activation reduces tumor growth and proliferation by inducing p21 and p27 expression and by downregulating the activity of cyclin D1 and c-myc [42]. On the other hand, GR signaling is strongly involved in the onset of resistance to anti-androgen therapies in advanced PC [43]. In particular, GR enhanced cell proliferation via activation of the signal transducer and activator of transcription 5 (STAT5), regardless of androgen receptor (AR) signaling, in castration-resistant prostate cancer (CRPC) cell lines treated with dihydrotestosterone (DHT) [44]. Conversely, in non-small cell lung cancer (NSCLC) and pancreatic cancer (PaC) preclinical models, GR has been found to act primarily as a tumor suppressor gene. In particular, in NSCLC cell lines treated with DEX, GR activation acted synergistically with p53 to promote cell cycle arrest [10]. Similarly, DEX administration resulted in PaC cells growth and EMT inhibition by suppressing NF-kB, IL-6 and vascular-endothelial growth factor (VEGF) [45]. From a clinical perspective, GR clinical activity should be considered in light of the wide spectrum of immunotherapy indications because GR signaling may exert a deleterious effect on the immune response [46]. Nevertheless, understanding the real impact of GCs on immunotherapy is challenging, in light of several factors, such as dose, timing, and therapeutic indications that may impact immunotherapy efficacy [46].

## 4. The Role of GR in Ovarian Cancer

The role of GR-mediated signaling in OC carcinogenesis and its impact on treatment response are controversial (Figure 2). An association between GR expression and clinical outcome in patients diagnosed with OC has been reported, which suggests that GR activity negatively affects tumor progression [47]. A study of 481 OC patients investigated the relationship between GR protein expression, evaluated by immunohistochemistry (IHC) on tumor samples, and clinical outcomes such as progression-free survival (PFS) and overall survival (OS) [47]. Notably, median PFS was significantly shorter in patients with a higher protein expression of GR (IHC 2+ or 3+) versus those with a low GR expression (IHC 0 or 1+), 20.4 versus 36 months, respectively. No significant correlation between GR expression levels and OS was observed [47]. In addition, the magnitude of benefit appeared to be strictly dependent on OC histotype since there was a substantial correlation between GR expression and PFS in low-grade and non-serous OC but not in HGSOC histology on the main results [47]. The association between GR activity and poor outcomes in OC could be partially explained by GC impairment of chemotherapy efficacy. GCs are widely prescribed to mitigate the most common side effects of chemotherapy such as chemotherapy-induced nausea and vomiting (CINV) and hypersensitivity reactions (HSRs) [48,49]. Additionally, GCs are largely prescribed in advanced stages since they are recommended by several international guidelines for the management of pain, asthenia, and cachexia [50,51,52]. Remarkably, platinum-based drugs and taxanes, which represent the backbone of the chemotherapy regimen adopted during OC treatment, are usually associated with emesis and HSRs and frequently require GCs administration. Notably, GC axis signaling has been shown to interfere with chemotherapy efficacy by reducing the apoptotic pathways enhanced by cytotoxic agents in both clinical and preclinical studies [53,54]. Of note, DEX administration induces chemoresistance in OC cell lines (OVM, M130, OAW-42, SKOV-3) treated either with cisplatin or gemcitabine. In particular, the administration of DEX 48 h prior to chemotherapy, resembling peak and basal plasma levels commonly found in vivo, prevented apoptosis induced by chemotherapy [53]. In addition, DEX treatment promoted OC cell proliferation and increased the expression of MKP-1 and SGK-1 proteins [54]. In nude mice injected with primary OC cells, the addition of DEX to chemotherapy increased the rate of tumor growth compared with chemotherapy alone [55]. GR activation has been found to inhibit the programmed cell death cascade by directly activating the transcription of several genes such as SGK1 and MKP1/DUSP1 [55]. SGK1 is a serine/threonine kinase that shares highly homologous sequences with AKT and that is activated by the PI3K signaling cascade. SGK1 has been suggested to prevent the expression of genes involved in apoptotic processes by phosphorylating Forkhead transcription factors such as Forkhead1 (FKHRL1) [56]. The inhibition of SGK1 counteracts the development of paclitaxel resistance and restores chemotherapy sensitivity in xenografted OC cell lines (A2780) by modulating the expression of Ran-specific binding protein-1 (RANBP1), which is an enzyme required for mitotic spindle-assembly and mitosis progression [57]. On the other hand, MKP1/DUSP1 gene encodes for a MAPK phosphate that prevents apoptosis by inhibiting the p38MAPK and c-Jun N-terminal kinase (JNK) transduction pathways [58,59]. Apoptosis impairment upon GR axis activation has been found in clinical samples as well. IHC MKP1 expression has been investigated in OC patients and is correlated with survival outcomes [60]. Strong to moderate MKP1 expression levels were found to be significantly associated with shorter PFS. In detail, mPFS was 18.3 vs. 40.6 months in OC patients with MKP1-positive and MKP1-negative tumors, respectively [60]. Furthermore, a randomized, placebo-controlled trial evaluated the expression of antiapoptotic genes in tumor specimens positive to GR expression (GR+) derived from patients with OC randomized to receive DEX or normal saline (NS) administration [60]. SGK1 and MKP1 mRNA levels were determined by reverse transcription polymerase chain reaction (RT-PCR) in OC tumor samples collected during exploratory laparotomy. The average expression of SKG1 and MKP1 mRNA was higher in patients receiving DEX than placebo [61] (Table 1). In addition, in preclinical models, DEX has been shown to induce chemoresistance by promoting ROR1 expression [62]. ROR1 activation induced by DEX promoted a stemness phenotype and resistance to platinum and taxane chemotherapy agents via the upregulation of diverse components of intracellular signaling pathways, including Ras homolog family member A (RhoA) and yes-associated protein 1/transcriptional coactivator with PDZ-binding motif (YAP/TAZ) [62]. In addition, a marked expression of ROR2 has been observed in platinum-resistant OC cell lines and is associated with upregulation of Wnt family member 5Aa (Wnt5a), signal transducer and activator of transcription 3 (STAT3) and NF-kB levels [63]. Furthermore, GR-mediated signaling interferes with OC cell adhesion and dissemination. DEX significantly increased the expression of α4β1, α5β1, integrin, and fibronectin, thereby resulting in increased cell adhesion to the extracellular matrix (ECM) and resistance to chemotherapy in SKOV-3 and HO-8910 OC cell lines [64]. DEX has been shown to promote the expression of transforming growth factor-β (TGF-β) type II receptor and to synergistically act with TGF-β1 secreted by ovarian epithelial cells to enhance OC cell adhesion and resistance to chemotherapy [64]. Moreover, in OC cell lines (PEO-14, SKOV-3), GR-mediated signaling has been shown to negatively regulate the expression of secreted Slit glycoproteins (SLITs)/roundabout receptors (ROBOs) that have been described as candidate tumor suppressor genes in OC [65]. In contrast, GR signaling has been reported to hamper OC metastasis by inducing microRNA (miRNA) expression. In SKOV-3 OC cells, low-dose DEX treatment induced the expression of miR-708 [66] that has been suggested to act as a tumor suppressor by significantly inhibiting cell invasion and dissemination. MiR-708 interferes with the GTPase Ras-proximity-1B (Rap1B) protein, whose activity has been shown to specifically promote epidermal growth factor receptor (EGFR)-dependent cancer cell migration [67,68]. In addition, the biological role of miR-708 has been described in mice bearing ID-8 cell-derived ovarian tumors treated with low-dose DEX [69]. Consistent with previous findings, DEX administration reduced primary tumor growth and abdominal metastasis via miR-708 upregulation and tumor microenvironment reprogramming [66]. Notably, a statistically significant association between higher miR-708 expression and improved survival rates, in terms of OS and RFS, has been observed in patients with advanced OC [66]. DEX decreased the expression of proinflammatory cytokines such as IL-1β and IL-18 and suppressed the recruitment of immune suppressive tumor-associated-macrophages (TAMs) and myeloid-derived suppressor cells (MDSCs) [70,71]. TAMs promote OC cells invasion and infiltration by activating NF-kB and c-Jun NH2-terminal kinase II (JNKII) and by inducing a chemokine gradient that eventually favors regulatory T-cell (T-regs) recruitment in the OC microenvironment [72,73]. Moreover, MDSCs have been shown to promote tumor immune escape by enhancing OC cell stemness through activation of the CSF2/p-STAT3 signaling pathway [74], and high MDSC levels have been independently associated with poor outcomes in OC patients [75].

## 5. GR and BRCA

GR activity appears to be strictly dependent on the mutation status of the breast cancer gene (*BRCA*) as functional crosstalk occurs. BRCA1/2 proteins act as crucial interactors in the homologous recombination (HR) mechanism, which is one of two major pathways of the DNA double-strand break (DSB) repair system during the S and G2 cell cycle phases [77,78]. BRCA pathogenic variants occur in up to 20% of HGSOCs [79] and have been associated with increased sensitivity to platinum-based chemotherapy [80] and PARP inhibition [81]. Remarkably, *BRCA1* has been shown to interact directly with nuclear steroid receptors in BC models, thereby acting as a repressor and a co-activator of ERα and AR transcriptional activity, respectively [82,83]. In addition, loss of GC activation has been shown to be dependent on BRCA mutation in TNBC tissues and in HR+ BC cell lines [84]. In particular, *BRCA1* loss of function was associated with reduced GR IHC expression in TNBC samples, whereas in HR+ MCF-7 cells, genetic inhibition of *BRCA1* decreased GR mRNA levels [84]. Moreover, *BRCA1* seems to enhance GR activity by modulating MAPK signaling. *BRCA1* appeared to induce the phosphorylation of various kinases, including p38, which promotes GR auto-transcription and GR-dependent gene expression [84]. Similarly, GR, in the absence of its ligand, positively regulates *BRCA1* expression in BC cells by directly interacting with the β-subunit of the transcription factor GA-binding protein (GABP) and the *BRCA1* promoter region [83]. Conversely, upon GC binding, GR loses its positive regulatory effect, thereby supporting a potential link between GR-mediated stress response and cancer development [83]. A *BRCA1*-GR interaction has also been observed in OC patients as well [84]. The effect of *BRCA1* on GR expression has been evaluated on 146 serous OC samples collected between 2010 and 2012 [85]. *BRCA1* mutated OC exhibited dramatically reduced GR protein expression versus *BRCA1* wild-type tumors, whereas a positive correlation between GR and BRCA1 was observed in BRCA1 wild-type samples.

*BRCA1* knockdown in OC cells (SKOV3) resulted in decreased GR mRNA levels in TNBC specimens [84,85]. Although transcriptional crosstalk may exist between GR and *BRCA1*, the impact on the clinical outcome of OC patients remains unclear. An analysis of 222 serous OC specimens selected from the Cancer Genome Atlas, collected from newly diagnosed OC patients, showed that high *NR3C1* gene expression appeared to be independently associated with decreased OS, regardless of BRCA mutation status [76]. The poorer outcomes were observed among *BRCA1* wild-type patients with higher *NR3C1* levels, thus suggesting that GR-mediated signaling may affect chemotherapy response independently of DNA-damage repair system defects [76]. However, although controversial, the bidirectional interplay between GR and BRCA may represent a new synthetic lethal interaction that may be explored to identify novel therapeutic targets and biomarkers for patient selection.

## 6. Glucocorticoid Receptor as a Potential Target for a Therapeutic Intervention in Ovarian Cancer

Since GR signaling plays a key role in resistance to chemotherapy, GR has emerged as a potential therapeutic target to improve chemotherapy efficacy in OC patients. However, current data regarding the major therapeutic options available may appear controversial and contradictory. Treatment activity and patient selection should be analyzed in light of multiple variables, such as GR and BRCA mutational status. Primarily, great research efforts have focused on the development of selective and non-selective GR antagonists that can antagonize cortisol activity and restore sensitivity to chemotherapy (Figure 3). The synthetic steroid mifepristone (MF) that acts as a progesterone receptor (PR) and GR antagonist effectively inhibits OC cell proliferation in preclinical models [86]. MF significantly reduced tumor growth regardless of PR expression by promoting nuclear localization of p21 and p27, inhibition of cdk2 activity and G1 phase arrest in OC preclinical models [86,87]. In addition, OC cell lines (SKOV-3) treated with MF exhibited a considerable impairment of adhesion, invasion, and metastatic potential [86,88]. In particular, MF administration induces cytoskeletal remodeling and nuclear distribution of fibrillar actin, thereby impairing adhesion capacity and reducing OC cell invasiveness [88]. Moreover, in an organotypic model system, MF inhibited HGSOC cell adhesion and promoted dissociation from the mesothelial cell monolayer [89].

Despite its promising preclinical activity, the efficacy of MF showed controversial results in OC clinical trials. In a single-arm phase II trial conducted in 44 OC platinum-resistant patients, daily oral MF exerted significant activity with an overall response rate (ORR) of 26.5%: three complete responses (CR) (9%) and six partial responses (PR) (17.5%) [90]. On the contrary, a subsequent multi-institutional phase II trial evaluating MF in platinum-resistant HGSOC patients reported a response rate of only 4.5% [91]. Patients in both trials were not stratified according to GR expression, which partially justifies the contradictory results. Furthermore, preclinical studies conducted on OC transgenic mice and cultured human HGSOC cells have shed light on the potential biphasic role of MF in OC progression [92]. The growth inhibitory effect on OC cell proliferation, largely described in in vitro models, has rarely been detected in vivo, probably because of the rapid metabolism and the high plasma protein binding rate of MF [92]. In OC human and murine cell lines, MF administration resulted in promoting cell proliferation by stimulating progesterone receptor membrane component 1 (PGRMC1) independently from GR signaling [92]. 

A new promising compound is overcoming the limitations and controversies associated with nonselective GR antagonists. Relacorilant (RELA) is a selective GR modulator (SGRM) that, unlike MF, does not bind PGR. SGRMs are novel emerging molecules that can act as both agonist as well as antagonist factors. Consequently, they target selected downstream signaling pathways, depending on the recruitment of specific coregulators [21]. SGRM activity depends on the levels of tissue-specific GR coactivators or corepressors similarly to the selective estrogen receptor modulator, Tamoxifen [93,94]. Moreover, SGRMs promote GR transrepression, which directs the anti-inflammatory effects of GCs, thereby reducing those adverse effects associated with GR transactivation, such as osteoporosis, hyperglycemia, and myopathies [95]. Several compounds belonging to this pharmacological class have been identified, both synthetic and natural, as described elsewhere [96]. RELA increased the potency and the cytotoxic activity of microtubule-targeted agents such as paclitaxel and gemcitabine by restoring apoptotic pathways suppressed by cortisol in OC cell lines (OVCAR5) and in xenograft models [97]. RELA combined with cytotoxic agents enhanced cancer cell viability, thereby increasing drug potency and efficacy, whereas no significant effect on tumor growth was observed with RELA monotherapy. The efficacy and safety of RELA was also evaluated in a phase II trial (NCT05257408) that enrolled patients diagnosed with recurrent platinum resistant or refractory HGSOC, endometrioid OC or ovarian carcinosarcoma who had received up to four chemotherapeutic regimens [98]. These patients were randomized to intermittent or continuous oral RELA plus nab-paclitaxel versus nab-paclitaxel monotherapy. The primary PFS analysis, after 11.1 months of median follow-up, confirmed a statistically significant benefit of 1.8 months for intermittent RELA plus nab-paclitaxel arm versus nab-paclitaxel alone (5.6 vs. 3.8 months) **[98]**. In addition, an updated survival analysis at 22.5 months of median follow-up showed a median OS of 13.9 months in the intermittent RELA arm compared with 12.2 months in the nab-paclitaxel arm, with low TGF-β1 levels found to be predictive of longer OS [99]. Treatment benefit was mainly observed among patients with high tumor GR expression who most benefited from adding RELA to nab-paclitaxel treatment (ORR = 40.4% vs. 18.8%). In addition, continuous or intermittent RELA administration resulted in significant suppression of canonical GR target genes expression such as SGK1, phosphatidylinositol-4,5-Bisphosphate 3-Kinase catalytic subunit gamma (PIK3CG), and glycogen synthase kinase 3 beta (GSK3B) compared to the nab-paclitaxel arm treatment, thus suggesting pleiotropic activity of GR antagonist [98,99]. Regarding the safety profile, the most frequent grade ≥ 3 adverse events were neutropenia and anemia peripheral sensory neuropathy in the intermittent RELA arm [98,99]. A phase III randomized trial (ROSELLA, NCT05257408) is currently evaluating the intermittent RELA plus nab-paclitaxel schedule vs. investigator’s choice of chemotherapy in platinum-resistant or refractory HGSOC who had received up to 1–3 prior lines of chemotherapy and at least one platinum-based regimen [100] (Table 2). 

GR pharmacological inhibition is a promising therapeutic strategy for OC treatment. However, the activation of GR signaling induced by DEX may also positively influence treatment efficacy. DEX-induced GR activation has been shown to increase sensitivity to several AKT/PI3K targeted kinase inhibitors by enhancing AKT phosphorylation in OC cell lines [62]. Conversely, DEX-mediated signaling, which upregulates ROR1 expression and directly affects tumor cell apoptosis, appeared to reduce the efficacy of secondary mitochondria-derived activator of caspase (SMAC) mimetics that counteract inhibitors of apoptosis that are highly expressed and dysregulated in OC [62,101]. However, the growing interest in GR inhibition collides with GCs’ broad therapeutic indications in cancer patients such as treatment of cancer-related fatigue (CRF), cachexia, and CINV. Nevertheless, alternative therapeutic options are available to treat these clinical conditions. Among these, physical exercise reduces CRF, thereby ensuring treatment adherence, and progestins stimulate appetite and reduce the release of cytokines associated with cachexia [52,102]. In addition, neurokinin-1 (NK-1) receptor antagonists and second-generation 5-HT3 receptor antagonists are efficacious options to CINV prophylaxis and treatment as an alternative or combined with GCs in order to safely reduce the minimum effective dose of GCs needed [48].

## 7. Conclusions

GR is a ligand-dependent transcription factor that plays a controversial role in OC biology. Several molecular mechanisms have been proposed to explain the impact of GR-mediated signaling on tumor development and progression, but uncertainties remain. GR is an independent prognostic factor in patients with OC since higher expression of GR in OC tissues correlates with resistance to chemotherapy and poor outcomes. Conversely, low doses of GCs significantly suppress OC growth and metastases in preclinical models by enhancing immune microenvironment remodeling, thereby directly reducing cancer cell migration. In addition, transcriptional crosstalk between BRCA1 and GR exists, although its impact on tumor phenotypes and clinical outcomes has not yet been completely clarified. Novel promising therapeutic strategies such as RELA have emerged to modulate GR signaling, to enhance chemotherapy sensitivity, and finally to provide novel effective treatment options for heavily pre-treated chemotherapy-resistant OC patients. 

## Figures and Tables

**Figure 1 biomolecules-13-00653-f001:**
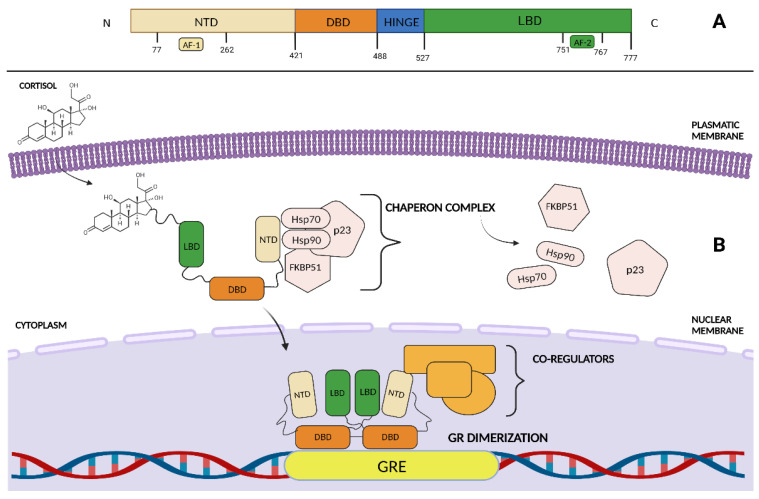
(**A**) The glucocorticoid receptor (GR) linear domain structure. NTD: amino-terminal domain. DBD: DNA-binding domain. LBD: ligand-binding domain. AF-1: activation function domain 1. AF-2: activation function domain 2. (**B**) In the absence of ligand, GR associates with chaperones in the cytosol. Upon GCs binding, GR translocates into the nucleus and interacts as a dimer with both DNA response elements and co-regulators to enhance gene expression. Hsp 90: heat-shock protein 90, Hsp70: heat-shock protein 90, FKBP51: FK506-binding protein 51. GRE: glucocorticoid responsive elements. Created with BioRender.com.

**Figure 2 biomolecules-13-00653-f002:**
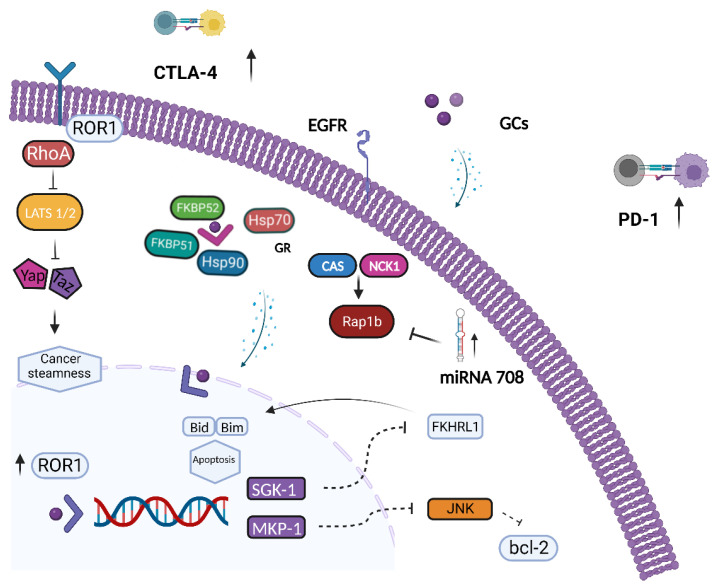
Upon cortisol binding, the glucocorticoid receptor enhances SGK-1 and MKP-1 expression to suppress ovarian cancer cell apoptosis. SGK1 has been hypothesized to phosphorylate FKHRL1, leading to its cytoplasm translocation, thereby preventing BID and BIM genes expression. The MKP1/DUSP1 gene encodes for a MAPK phosphate that preferentially inhibits p38MAPK and JNK transduction pathways, thereby resulting in decreased anti-apoptotic protein BCL-2 levels. Moreover, glucocorticoid-mediated signaling induces ROR1 expression, which in turn promotes a cancer stemness phenotype via RhoA, YAP/TAZ, and BMI-1 downstream upregulation. Additionally, GR eventually facilitates cancer immune evasion by enhancing the expression of both PD-1 on T-cells and NK-cells and CTLA-4 on activated CD4+ and CD8+ T-cells. Conversely, GR signaling increases miR-708 levels, thus resulting in impaired OC cell dissemination through targeting EGFR downstream effector Rap1B. Created with BioRender.com.

**Figure 3 biomolecules-13-00653-f003:**
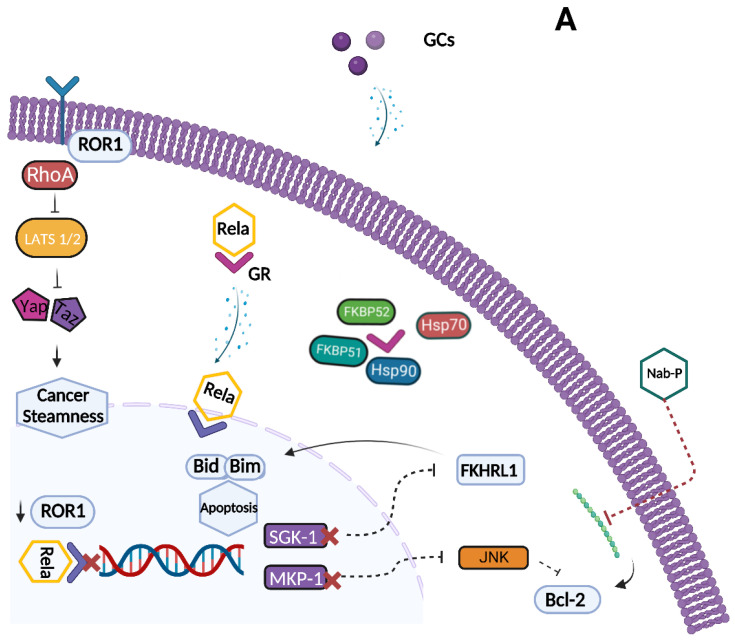
(**A**) Relacorilant is a selective GR modulator that restores the programmed cell death cascade that is suppressed by cortisol. Relacorilant acts by targeting MKP-1 and SGK-1 gene transcription. SGK1 has been hypothesized to phosphorylate FKHRL1, thereby leading to its cytoplasm translocation and preventing BID and BIM gene expression. The MKP1/DUSP1 gene encodes for a MAPK phosphate, which preferentially inhibits p38MAPK and the c-Jun N-terminal kinase (JNK) transduction pathways, thereby resulting in decreased anti-apoptotic protein BCL-2 levels. (**B**) Mifepristone is a synthetic steroid that acts as a GR antagonist. It reduces OC cell growth by promoting the nuclear localization of p21 and p27 and by decreasing cdk2 activity, thus resulting in OC cell G1 phase arrest. In addition, in preclinical models, MF promoted cytoskeletal rearrangements such as membrane ruffling, lacking adhesion capacity and nuclear distribution of fibrillar actin, reducing OC cell invasive capacity. Moreover, MF administration resulted in decreased PD-1 and CTLA-4 expression, which suggests that GR plays a role in tumor immune evasion. Nab-Paclitaxel is a cell cycle specific agent that prevents microtubule depolymerization and inhibits BCL-2 protein-enhancing cell apoptosis. Created with BioRender.com.

**Table 1 biomolecules-13-00653-t001:** Retrospective trials exploring GR activity.

Study Objective	Study Type	# of Pts	Results	References
Association between GR IHC expression and outcome of OC patients	Retrospective	481	mPFS = 20.4 months (GR IHC 2+, 3+) vs. 36 months (0 or 1+) HR = 1.66, 95% CI 1.29–2.14, *p* = 0.001	[47]
Association between NR3C1 gene expression and overall survival of OC patients	Retrospective	222	Trend toward decreased OS in pts with high NR3C1 expression compared with low NR3C1 expression (*p* = 0.06) independently from BRCA mutational status	[76]
MKP1 expression in primary human ovarian carcinoma	Retrospective	101	mPFS = 18.3 (MPK1+) vs. 40.6 months (MKP1-) (95% CI 13.11–23.5, *p* = 0.019)	[60]
Association between MiRNA-708 expression and OC patients’ survival	Retrospective	82	Pts with high miR-708 expression had a significantly better OS (*p* = 0.04) and RFS (*p* = 0.026) than those with low miR-708 expression	[66]
Expression of the anti-apoptotic genes SGK1 and MKP1/DUSP1 in ovarian tissues upon DEX or NS administration	Prospective randomized	10	The average SKG1 and MKP1 mRNA expression was increased 6.1-fold vs. 1.5 and 8.2-fold vs. 1.1, in the DEX and NS arms respectively compared with baseline pretreatment levels	[61]

CI, confidence interval; DEX, dexamethasone; GR, glucocorticoid Receptor; HR, hazard ratio; IHC, immunochemistry; mPFS, median progression-free survival; NS, normal saline; OC, ovarian cancer; OS, overall survival; pts, patients; RFS, recurrence-free survival.

**Table 2 biomolecules-13-00653-t002:** Results from selected phase II/III trials of systemic treatment targeting GR.

Study Objective	Phase	# of Patients	Results	References
Oral MF activity in refractory EOC	II	44	ORR = 26.5%, CR = 9%, PR = 17.5%	[90]
MF activity in recurrent or persistent HGSOC	II	24	Only 1 patient had a partial response (4.5%)	[91]
RELA efficacy and safety in combination with chemotherapy in Platinum-Resistant HGSOC	II	60 (intermittent RELA plus Nab-P) vs. 58 (continuous RELA plus Nab-P) vs. 60 (Nab-P)	mPFS = 5.6 months vs. 3.8 months, HR 0.66, 95% CI 0.44–0.98, *p* = 0.038); ORR High GR expression = 40.4% vs. 18.8% χ *p* 0.037; ORR low GR expression= 32.0% vs. 47.1% χ *p* > 0.05, OS = 13.9 months vs. 12.2 months HR 0.63, *p* = 0.045.	[98,99]
Intermittent RELA plus nab-paclitaxel efficacy vs. TPC in in platinum resistant pre-treated HGSOC (ROSELLA, NCT05257408)	III	360 (estimated)	Ongoing	/

Abbreviations: CI, confidence interval; CR, complete response; EOC, epithelial ovarian cancer; GR, glucocorticoid receptor; HGSOC, high-grade serous ovarian cancer; HR, hazard ratio; MF, mifepristone; mPFS, median progression-free survival; Nab-P, nab-paclitaxel; ORR, overall response rate; PR, partial response; pts, patients; RELA, relacorilant; TPC, treatment of physician’s choice.

## Data Availability

Not applicable.

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
