# Peer review of "Glucocorticoid Receptor and Ovarian Cancer: From Biology to Therapeutic Intervention"

_biomolecules, 2023, doi:10.3390/biom13040653_

Round 1
Reviewer 1 Report
Dear Authors,
Thank you for the opportunity to review your paper. The subject you wrote about is interesting, new and not very much studied. I have three minor suggestions:
1. To add in the title one or two words that describe what type of paper is your publication- like " narrative review"
2. To complete at the end of the article the section " author contributions"
3. On the "abstract" to ad one or two sentence about the conclusion of your paper.
Author Response
Reviewer #1 Dear Authors, Thank you for the opportunity to review your paper. The subject you wrote about is interesting, new and not very much studied. I have three minor suggestions:
Q1. To add in the title one or two words that describe what type of paper is your publication-like "narrative review".
A1. Thank you for the suggestion. The title now reads as follows (line 1 of revised manuscript): Narrative Review. Glucocorticoid Receptor and Ovarian Cancer: From Biology to Therapeutic Intervention.
Q2. To complete at the end of the article the section "author contributions"
A2. (Lines of revised manuscript 513-515)
Author Contributions: conceptualization, R.B., G.N, and C.D.A.; methodology, C.D.A., G.A., M.G., P.D.P., E.P.; investigation, R.B., G.N., A.C., A.C., S.C.C.; resources, C.D.A., G.A., M.G., E.P., P.D.P.; writing—original draft preparation, R.B., G.N.; writing—review and editing, C.D.A., G.A., M.G.; visualization, R.B., G.N.; supervision, C.D.A.
Q3. On the "abstract" to add one or two sentence about the conclusion of your paper.
A3. (Lines of revised manuscript 27-33)
Abstract: Ovarian cancer (OC) is the leading cause of death from gynecological malignancies worldwide. Fortunately, recent advances in OC biology and the discovery of novel therapeutic targets led to the development of novel therapeutic agents that may improve the outcome of OC patients. The glucocorticoid receptor (GR) is a ligand-dependent transcriptional factor known for its role in body stress reactions, energy homeostasis and immune regulation. Notably, evidence suggests that GR may play a relevant role in tumor progression and may affect treatment response. In cell culture models, administration of low levels of glucocorticoids (GCs) suppress OC growth and metastasis. Conversely, high GR expression has been associated with poor prognostic features and long-term outcomes in patients with OC. Moreover, both preclinical and clinical data have shown that GR activation impairs the effectiveness of chemotherapy by inducing the apoptotic pathways and cell differentiation. In this narrative review, we summarize data related to the function and role of GR in OC. To this aim, we reorganized the controversial and fragmented data regarding GR activity in OC and described its potential use as a prognostic and predictive biomarker. Moreover, we explored the interplay between GR and BRCA expression and reviewed the latest therapeutic strategies such as non-selective GR antagonists and selective GR modulators to enhance chemotherapy sensitivity, and to finally provide new potential options in OC patients.
Reviewer 2 Report
1. The manuscript suffers from lack of careful proofreading, in some places rather severely. It is recommended that the authors have their manuscript proofread by someone proficient in English.
2. Many statements should be referenced that are not. For example, “Notably, GR expression has been associated with a favorable prognosis in estrogen-receptor positive (ER+) breast cancer (BC).” Please make sure such important statements are referenced.
3. Some statements that are bolded (bold font) in the text, and it is not clear why.
4. There is no methods section. How were searches conducted? What were the inclusion / exclusion criteria for article acquisition? What type of review is this (it’s not a systematic review)?
5. There is no limitations section, or a section on future directions.
6. The manuscript is written as a long series of findings, often crammed into very long paragraphs. This can make for tedious reading as presented, without corresponding figures or much interpretation. Section 4, The Role of GR in Ovarian Cancer is, of course, a very important one. Here, at least, I would highly recommend that the authors tabulate important data so that the reader can keep track of and compare important findings. Information in this paragraph jumps from human studies to in vivo studies, to in vitro studies. At least the human data could be tabulated, and I would also consider tabulating the results of preclinical models as well. I think it will be very difficult for a typical reader to keep track of all the findings as they are currently presented.
7. Section 6, Glucocorticoid receptor as a potential target for therapeutic intervention in ovarian cancer, might also benefit from the tabulation of key study findings. It may otherwise be difficult for readers to follow.
Reviewer 3 Report
This work provided an overview insight the major evidences related to the function and role of GR in ovarian cancer. The authors described not only the physiological and pathological functions of GR, but also the potential use-fulness of GR as a prognostic and predictive biomarker and the latest therapeutic strategies emerging to modulate its activity. The manuscript is in general well written and the relevant research progress on the relationship of GR and ovarian cancer is described in detail. However, the following issues have to be addressed before this manuscript is suitable for publication.
1. In the section 2. GR Structure and Function, the authors mentioned that GR belongs to the nuclear receptor superfamily. More information about the nuclear receptors, not just GR, should be introduced herein. The following references are suitable to be cited.
https://doi.org/10.1016/j.fct.2022.113462
https://doi.org/10.1016/j.fct.2022.113407
2. In the section 6. Glucocorticoid Receptor as a Potential Target for Therapeutic Intervention in Ovarian Cancer, line 361, a selective GR modulator (SGRM) relacorilant was introduced. However, the background information about SGRMs was missing. What about the special functions of SGRMs? What is the difference between natural and synthetic SGRMs? The authors should discuss more deeply.
https://doi.org/10.1016/j.phrs.2020.104802
https://doi.org/10.1073/pnas.1219411110
3. Please improve the clarity of Fig. 3.
4. The novelty and innovative potential of your manuscript compared to the published literature should be described in more detail in the abstract and discussion section.
Round 2
Reviewer 2 Report
It's still unclear how the authors selected literature for review to avoid cherry-picking and provide readers with a complete, unbiased picture. There is no methods section, and no limitations section. Also, the English has improved, but still suffers.
Author Response
Dear Reviewer,
Thanks for your suggestions. As we previously explained, we mistakenly used the template for “Systematic review” instead of “Narrative review” during the submission process of our manuscript. To provide and extensive and complete overview of the biological and clinical data on the role of glucocorticoid receptor in ovarian cancer we conducted a literature search using the items “ovarian cancer”, “glucocorticoid receptor”, “chemotherapy resistance and ovarian cancer” in Pubmed, Embase and Google Scholar databases. Relevant papers published between January 1st 2000 and 1st December 2022 were identified and evaluated for inclusion in our review. Articles and abstracts published in the ESMO (European Society of Medical Oncology), ASCO (American Society of Clinical Oncology), and AACR (American Association for Cancer Research) Congresses between 2000 and 2022 were also reviewed. Full papers and presentations/abstracts reporting original data from pre-clinical and clinical research concerning the role of GR and glucocorticoid metabolism-targeting therapy in OC were assessed for inclusion in the Review. The most updated data were considered in case of duplicate publications.
We also requested a second proofreading of the manuscript to improve the English-language
We hope that these changes and responses are satisfactory.
Thanks,
Carmine De Angelis
